# Experience, Knowledge, and Perceptions of Pharmacogenomics among Pharmacists and Nurse Practitioners in Alberta Hospitals

**DOI:** 10.3390/pharmacy10060139

**Published:** 2022-10-26

**Authors:** Meagan Hayashi, Chad A. Bousman

**Affiliations:** 1Pharmacy Services, Alberta Health Services, Edmonton, AB T6G 2R3, Canada; 2Faculty of Pharmacy and Pharmaceutical Sciences, University of Alberta, Edmonton, AB T6G 2R3, Canada; 3Departments of Medical Genetics, Psychiatry, Physiology & Pharmacology, Community Health Sciences, University of Calgary, Calgary, AB T2N 1N4, Canada; 4Alberta Children’s Hospital Research Institute, University of Calgary, Calgary, AB T2N 1N4, Canada; 5Mathison Centre for Mental Health Research and Education, Hotchkiss Brain Institute, University of Calgary, Calgary, AB T2N 1N4, Canada

**Keywords:** pharmacogenomics, pharmacogenetics, hospital pharmacy, pharmacist, nurse practitioner, physician, healthcare

## Abstract

Background: Despite evidence of clinical utility and the availability of prescription guidelines, pharmacogenomics (PGx) is not broadly used in institutional settings in Canada. To inform future implementation, this study aimed to identify healthcare provider knowledge, experience, and perceptions of PGx in Alberta, Canada. Methods: An online 44-item survey was distributed to pharmacists, nurse practitioners, and physicians employed or contracted with Alberta Health Services from January to May 2022. Questions included: demographics, professional history, PGx education and exposure, knowledge, and ability to use PGx, and attitudes towards, feasibility, clinical utility, education, and implementation. Results: Ninety-one pharmacists, 37 nurse practitioners, and 6 physicians completed the survey. Fifty-nine percent had 10 or more years of experience, and 71% practiced in urban settings. Only one-third had training in PGx, and one-quarter had used PGx. Most respondents (63%) had no knowledge of PGx resources, including the Pharmacogenomics Knowledge Base (75%), or the Clinical Pharmacogenetics Implementation Consortium guidelines (85%). While participants agreed that they understood genetic (75%) and PGx (63%) concepts, most disagreed with their ability regarding practical applications of PGx such as translating genotype to phenotype (74%) or counselling patients on results (66%). Participants agreed on the clinical utility of PGx in preventing adverse drug reactions (80%) and enhancing medication efficacy (77%), and identified oncology (62%), cardiovascular/stroke (60%), and psychiatry (56%) as therapeutic areas to consider implementation. At present, healthcare provider knowledge (87%), cost (81%), and limited guidelines/evidence (70%) are seen as the greatest barriers to implementation. Conclusion: Alberta healthcare providers have limited training, experience, or knowledge in PGx. However, most appear to have a positive outlook regarding clinical utility, especially within oncology, cardiology, and psychiatry. More effort is required to socialize the availability and quality of evidence and guidelines for the interpretation of PGx test results, address other knowledge gaps, and improve financial limitations.

## 1. Introduction

Pharmacogenomics (PGx) uses genetic information to predict interindividual variability in pharmacokinetic and pharmacodynamic responses to medications [1,2]. PGx research to date has contributed to the development of prescribing guidelines by expert groups such as the Clinical Pharmacogenetics Consortium (CPIC) [3], the Dutch Pharmacogenetic Working Group (DPWG) [4], and drug labels applied to medications by the United States Food and Drug Administration (FDA) [5]. These guidelines have facilitated clinical implementation of PGx information globally [6] but, within Canada, the utilization of pharmacogenomics outside of the oncologic setting is sparse. A recent scoping review of pharmacist-implemented pharmacogenomics showed that only 9.3% of the 43 included studies took place within Canada [7]. The rationale for this may be multifactorial, with one consideration being healthcare provider familiarity and understanding of pharmacogenomics. In Alberta (a province in Western Canada), numerous efforts to implement PGx into routine clinical are underway. To inform Alberta’s implementation strategy and determine future readiness to implement pharmacogenomics, this study surveyed healthcare providers practicing in Alberta hospitals and institutional healthcare settings to identify their current knowledge, experience, and perceptions of PGx. 

## 2. Methods

### 2.1. Survey Design

A brief (~15 min) anonymous, cross-sectional, web-based survey with 44 items was developed using REDCap electronic data-capture tools hosted at the University of Alberta [8,9]. The design process involved a literature review of previous studies evaluating knowledge and opinions of PGx by pharmacists, physicians, nurses, and other health professions [10,11,12,13]. The questionnaire was divided into five sections: (1) Eligibility Questions (3 items); (2) Demographics and Professional History (4 items); (3) Pharmacogenomics Background (10 items); (4) Confidence & Self-Rated Knowledge (11 items); (5) Attitudes—Feasibility and Utility (6 items); (6) Attitudes—Education and Implementation (10 items). Some questions contained sub-questions that appeared with branching logic if participants identified prior experience or education with PGx. A copy of the survey can be found in the Appendix A.

### 2.2. Participants

Between January 2022 and May 2022, invitations to participate were sent by Alberta Health Services’ (AHS) professional department leaders for pharmacists and nurse practitioners. Physicians were recruited through an invitation to participate in the AHS Office of Medical Affairs and Alberta College of Physicians and Surgeons newsletters. Any pharmacist, nurse practitioner, or physician employed with or providing contracted services to AHS in a hospital or other institutional setting was eligible for inclusion. This study was conducted according to the guidelines of the Declaration of Helsinki and approved by the University of Alberta Research Ethics Board (Pro00114758, 13 October 2021). All participants provided informed implied consent prior to completion of the survey.

### 2.3. Statistical Analysis

Data were summarized as mean ± standard deviation or *n* (%) as appropriate for the whole sample and for subgroup strata. Answers were summarized as *n* (%) for whole sample and for subgroup strata, with comparisons performed using the chi-square test and a *p* < 0.05 considered statistically significant. All analyses were performed using Stata version 17 (StataCorp, College Station, Texas, USA).

## 3. Results

### 3.1. Participants

Invitations were sent to 1150 pharmacists and 450 nurse practitioners employed with AHS (response rates 7.9% and 8.2%, respectively). A link to the survey was also included in physician newsletters from the College of Physicians and Surgeons of Alberta and the AHS Office of Medical Affairs, where 11,000 physicians were reached (response rate 0.1%). Respondents were primarily pharmacists (67.9%) or nurse practitioners (27.6%). Most respondents had at least 10 years of experience (59.4%), with 31.6% reporting more than 20 years of experience. Participants were largely located in urban settings (71.2%) and came from a wide variety of clinical specialties, identified in the summary of professional settings of survey respondents in Table 1. Study sample demographics are representative of similar results in previous studies for years in practice [14,15], and, for pharmacists, for the location of practice [14]. No prior research was found regarding practice specialties for these populations.

### 3.2. Pharmacogenomics Training and Experience

Most participants (68.8%) had no prior training in PGx (Figure 1A). Of the 39 participants with prior training, most received PGx education in their qualifying degree program (*n* = 22), self-study (*n* = 20), or in a conference setting (*n* = 18) (Figure 1B). There was no difference in prior education between those who had been practicing for more than 10 years compared with those who had been in practice for a shorter time (31.9% vs. 30.2%, respectively; *p* = 0.834). One-fourth (26%) of participants had used PGx in their practice (Figure 1C). These applications included: formulating a care plan (*n* = 26), providing education to patients (*n* = 18) or other healthcare professionals (*n* = 16), ordering pharmacogenomic tests (*n* = 16), or providing pre-test education (*n* = 5) (Figure 1D). Among participants with prior use of PGx, 43.5% (*n* = 17) reported experience with less than 10 patients, 15% (*n* = 6) with 10–50 patients, 5% (*n* = 2) with 51–100 patients, and 13% (*n* = 5) with more than 100 patients. Three respondents indicated prior experience that was not in direct patient care. Subgroup data on training and experience are further detailed in Appendix A.

### 3.3. Knowledge of Pharmacogenomics

#### 3.3.1. Resource Knowledge

Most survey respondents (62.9%) had no knowledge of any PGx-specific resources included in the survey (Figure 2). Of the queried resources, the greatest awareness was for PharmGKB, with 26% having heard of the resource, 13% indicating familiarity with it, and 4% having prior use of it. Only 21%, 16%, and 12% had heard of the Canadian Pharmacogenomics Network of Drug Safety (CPNDS), CPIC, and DPWG guidelines, respectively. The most frequently accessed resources for PGx information reported by participants (*n* = 114) were Lexicomp (35.1%), the Pharmacogenomics Knowledge Base (22.8%), Micromedex (15.8%), Clinical Pharmacogenetics Implementation Consortium guidelines (11.4%), and drug monographs (11.4%). The majority of participants (58.8%) responded that they did not know where to look for PGx information.

#### 3.3.2. Applying Knowledge

While 74.8% and 62.9% of participants agreed that they understood basic genetic and pharmacogenetic concepts, respectively, most disagreed that they could identify patients (60.5%) and medications (50.0%) suitable for testing, select appropriate PGx laboratories (76.6%), identify (62.9%) and communicate (70.2%) the risks of PGx testing, translate genotype to phenotype (74.2%), or counsel patients on their PGx test results (65.9%). Responses were more varied regarding participant ability to explain PGx to patients (41.1% agreement) and other healthcare providers (29.0% agreement). A breakdown of Likert-scale responses is summarized in Figure 3A.

#### 3.3.3. Obtaining Knowledge

Participants indicated the strongest preference for self-study (*n* = 81, 68%), conference (*n* = 73, 61%) or certificate program (*n* = 68, 57%) methods of PGx education. On-site training (*n* = 49, 41%) and small group workshops (*n* = 39, 33%) were the least preferred. Fifteen survey respondents did not indicate a preferred method and were excluded from this analysis.

### 3.4. Perceptions of Pharmacogenomics

#### Clinical Utility

Most participants agreed with the clinical utility of PGx in clinical practice but were uncertain about its cost-effectiveness (Figure 3B). Oncology was consistently identified as the top clinical setting for pharmacogenomic implementation by survey respondents (Table 2). Other top areas included cardiology/stroke/vascular surgery, psychiatry, and general practice/multiple settings.

Pharmacists were consistently the primary practitioner selected to fulfill the duties of pre-test education, test interpretation, and post-test education in the patient care components of pharmacogenomic testing (Table 3). This was consistent between pharmacist and nurse practitioner respondents for pre-test education and test interpretation; however, regarding post-test education nurse practitioners were more varied in their responses vs. pharmacists. Due to the small sample size of physicians (*n* = 6), no subgroup analysis was performed with this group.

### 3.5. Barriers to Implementation

Lack of healthcare provider knowledge was largely considered the greatest barrier to pharmacogenomics’ implementation, followed by cost and lack of clinical guidelines (Table 4). Ethical, legal, and social considerations, as well as patient acceptance, ranked low among the barriers to pharmacogenomics implementation.

## 4. Discussion

PGx has become a staple of precision medicine due to its utility in medication selection and dosing with available evidence-based prescribing guidelines. PGx testing is used in institutions across the United States [16] and Europe [17]; however, implementation in Canada has greatly lagged behind these regions. A preliminary step to introducing a new evidence-based practice is to gather internal information and engage with stakeholders [18]. This research provides an understanding of the current landscape of PGx knowledge and attitudes among the front-line clinicians who would be eventual users of PGx in their clinical practice. Within Alberta, such practitioners are those with the ability to prescribe and/or assess medication: physicians, nurse practitioners, and pharmacists. Due to the low response rate from physicians, the results of this research are primarily representative of knowledge for the latter two groups. It was identified that pharmacists and nurse practitioners currently have limited knowledge and experience with PGx. Only 31% of respondents had any prior education or training in PGx, which is considered imperative to the implementation of pharmacogenomics into practice [19]. Unlike previous research that identified earlier-career practitioners as being more likely to have prior education in PGx [11,20], likely due to the relatively recent addition of PGx in entry-to-practice curricula [21], there was no such difference found in this sample. It is apparent that education will need to reach all current healthcare providers before PGx can be implemented, regardless of experience. This need for education is also demonstrated by the overall low-rated confidence that clinicians surveyed had in their ability to use and discuss PGx information in patient care activities. To address this knowledge gap, respondents identified a preference for self-study, conference, or certificate modes of education, similar to other research that was carried out in Canada [21]. Despite the limited knowledge, most respondents had a positive attitudes towards PGx’s ability to enhance medication efficacy and prevent adverse side-effects, with most agreeing that they desire to learn more about PGx and use it in their practice, and can see it being part of their practice in the next 10 years.

Owing to the low levels of reported training and exposure to PGx, there was also limited knowledge regarding the availability of PGx prescription information through the CPIC, DPWG, and PharmGKB. More than half of survey respondents reported that they do not know where to look for PGx information. This starkly contrasts with a survey of physicians in the United States, where PGx use is more prominent, in which only 14% responded that they had not consulted PGx resources in their practice [22]. Of those in Alberta that did indicate where they prefer to find PGx data, most used Lexi-Comp. While Lexi-Comp is useful in evaluating many factors in interpatient variability in drug response, such as organ function, drug interactions, and PGx, its PGx data are brief, limited, and do not always contain all available PGx information or CPIC guidelines [23]. PGx-focused resources such as PharmGKB improve the availability of PGx information; however, only a quarter of respondents indicated that they had heard of this resource. Ideally, clinicians should directly refer to the original PGx prescribing guidelines to ensure that only the most accurate information is used; however, only 11.4% indicated that they preferred using the CPIC guidelines for PGx information assessment when able to select more than one choice. Fewer than 20% of Alberta clinicians had heard of CPIC prior to this survey, indicating overall unawareness of PGx prescribing guidelines. Therefore, it is unsurprising that the largest perceived barrier to PGx implementation after clinician knowledge and cost is cited by respondents as lack of clinical evidence and guidelines. This research makes the case that the issue is not necessarily that there is insufficient evidence or guidance to use PGx in practice, but rather a lack of awareness of the current information that is available for clinicians to use in direct patient care activities. Prior implementation models in the United States support the concept of PGx resources embedded in the electronic medical record [24]. Such technology adaptations in PGx implementation may address the gaps the in provider knowledge of resources by providing information on-demand, and as some institutions have accomplished, education at the point of prescribing though “just-in-time” workflow learning [24,25]. With the current transition in Alberta to a new, province-wide electronic medical record in progress, PGx’s integration in direct care activities has become a more realistic possibility.

## 5. Limitations

The largest limitation of this research study is the lack of physician responses that were received (0.1% response rate). This research was conducted during the COVID-19 pandemic in Alberta, and after recent physician remuneration debates with local government. Therefore, physician burnout, survey fatigue, and competing priorities may all have played a role in the limited responses received from survey distribution through usual physician communication methods. Other research completed in Alberta a year earlier specifically targeting pediatricians and pediatric psychiatrists provides insight into this physician specialty [26]. As this survey identified similar rates of prior education, self-rated knowledge, and opinions on clinical utility as the pharmacists and nurse practitioners responding to the survey described in this manuscript, it is likely that the information provided by Jessel et al. [26] and this current study provide some insight into physician knowledge and attitudes in Alberta. This is a critical group for future PGx implementation, and thus it is important to further engage with other physician specialties in Alberta that have not been fully captured by the current and previous studies. Of particular importance are those practicing in oncology, cardiology, and geriatrics. This could be accomplished through focus groups or other qualitative methods, such as structured interviews, with more questions directed to the specialists’ area of practice and the specific drug–gene interactions found within.

Another limitation common with survey research include response bias, wherein only those with knowledge or experience with pharmacogenomics are inclined to respond to a survey about pharmacogenomics. However, the distribution of PGx knowledge and experience among participants in the current study align with previous surveys conducted in other jurisdictions, suggesting that response bias is unlikely to be of greater concern than in similar studies.

## 6. Conclusions

Alberta health care providers reported limited training, knowledge, and exposure to PGx information. Despite this, most health care providers appear to have a positive outlook towards the future applications of PGx, especially within oncology, cardiology, psychiatry, endocrinology, infectious diseases, geriatrics, and in general medicine practices. Findings from this study will inform future efforts in Alberta and beyond to socialize the availability and quality of evidence for the interpretation of PGx test results, and address barriers pertaining to knowledge gaps, costs, resources, and technology. However, further research is needed to identify the most cost-effective strategies for accomplishing these objectives to ensure the overall success of PGx implementation.

## Figures and Tables

**Figure 1 pharmacy-10-00139-f001:**
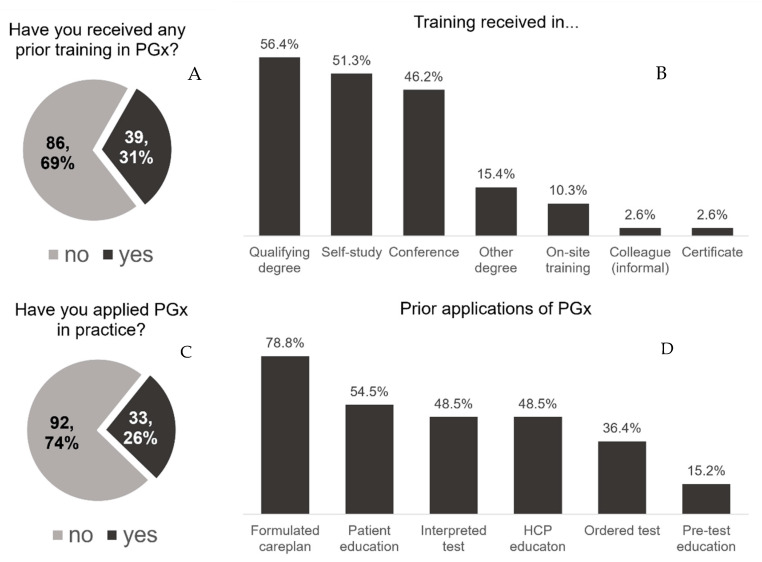
Prior training and uses in PGx. (**A**) Proportion of participants reporting prior training in PGx (*n* = 125). (**B**) Proportion of participants (*n* = 39) reporting receipt of training across seven mediums. Participants could select more than one medium to indicate where prior training was received. (**C**) Proportion of participants reporting use of PGx in practice, (**D**) Proportion of participants (*n* = 33) reporting utilization of PGx across six applications. More than one application could be selected. PGx: Pharmacogenomics, HCP: healthcare provider/professional.

**Figure 2 pharmacy-10-00139-f002:**
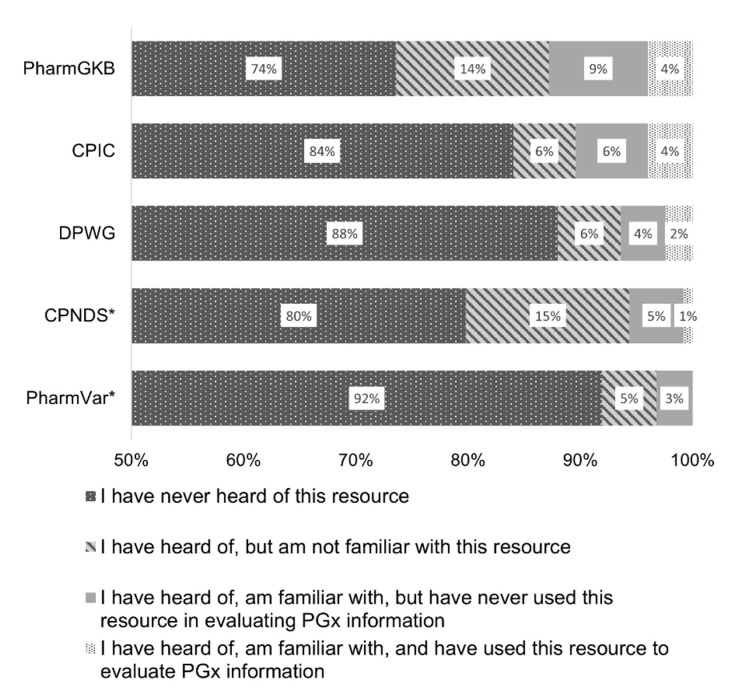
Awareness and use of pharmacogenomics resources among survey participants (*n*=125). * one missing response. PharmGKB: Pharmacogenomics Knowledge Base; CPIC: Clinical Pharmacogenetics Implementation Consortium; DPWG: Dutch Pharmacogenetics Working Group; CPNDS: Canadian Pharmacogenomics Network of Drug Safety; PharmVar: Pharmacogene Variation Consortium.

**Figure 3 pharmacy-10-00139-f003:**
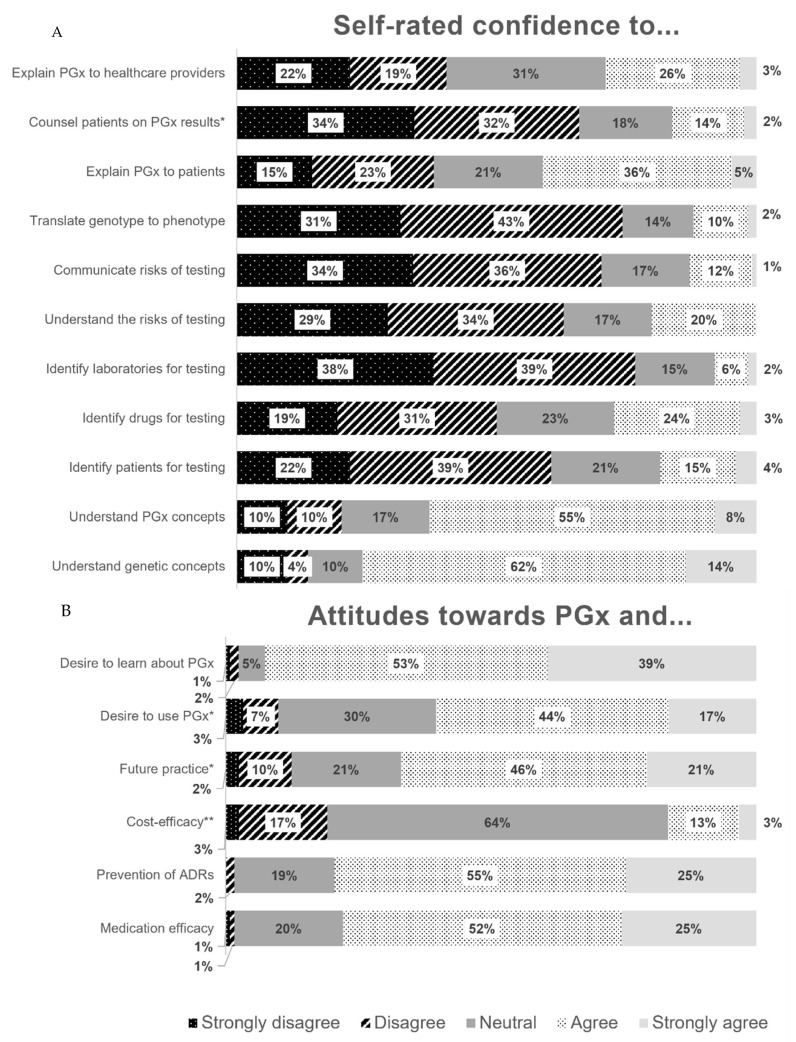
(**A**) Participant responses (*n* = 124) to Likert scale questions regarding self-rated confidence in knowledge and abilities in pharmacogenomics (PGx). (**B**) Agreement with statements regarding PGx’s utility and future in practice (*n* = 122). Full survey questions can be found in the Appendix A, survey Part 2 and Part 3A for confidence and attitudes, respectively. * one missing response; ** two missing responses.

**Table 1 pharmacy-10-00139-t001:** Demographics of survey respondents.

Characteristic		*n*	(%)
**Profession**			
	Pharmacist	91	(67.9)
	Nurse Practitioner	37	(27.6)
	Physician	6	(4.5)
**Years in Practice ***			
	Less than 2 years	8	(6.0)
	2–5 years	20	(15.0)
	6–10 years	26	(19.6)
	11–15 years	18	(13.5)
	16–20 years	19	(14.3)
	More than 20 years	42	(31.6)
**Location of Practice** **(Number of inhabitants) †**			
	Rural (0–50,000) or Locum	13	(9.8)
	Suburban (50,001–250,000)	25	(18.9)
	Urban (greater than 250,000)	94	(71.2)
**Specialty †**			
	None/general medicine	25	(18.9)
	Pediatric/neonatal medicine	17	(12.9)
	Oncology	17	(12.9)
	Adult intensive care or emergency medicine	17	(12.9)
	Psychiatry	10	(7.5)
	Cardiology and stroke	10	(7.6)
	Geriatrics	9	(6.8)
	Infectious diseases	7	(5.3)
	Pain and palliative care	7	(5.3)
	Other ‡	14	(9.8)

*n = 134* * one missing response † two missing responses; ‡ (non-cardiac surgery (*n* = 2), anesthesia (*n* = 1), human immunodeficiency virus (*n* = 1), neurology (*n* = 1), transplant (*n* = 1), physiatry (*n* = 1).

**Table 2 pharmacy-10-00139-t002:** Top therapeutic areas selected for implementation by survey respondents.

Therapeutic	% Selected * (*n* = 114)	% Selected as Top Therapeutic Area (*n* = 116)
Oncology	62.3	35.3
Cardiovascular/stroke/vascular surgery	59.7	7.8
Psychiatry	56.1	19.0
Endocrinology/diabetes	46.5	4.3
Infectious diseases	34.2	0
Geriatrics	34.2	2.6
No specialty/general practice	33.3	19.5
Pediatrics	29.0	4.3
Nephrology	28.1	0
Gastroenterology	27.2	0
Critical care	27.2	0.9
Respiratory medicine	25.4	0
Pain management and palliative care	19.3	0
Emergency medicine	9.7	0
Other **	4.4	n/a
Outpatient medicine	0	0.9
Unsure/unable to say	n/a	5.2

* Participants could select more than one response. ** neurology (*n* = 1), anesthesia (*n* = 1), human immunodeficiency virus (*n* = 1); unsure (*n* = 2).

**Table 3 pharmacy-10-00139-t003:** Respondents’ perceptions on optimal professions best-suited to implement pharmacogenomics.

Profession	Component of Patient Care with PGx *n* (%)		
	Pre-Test Education	Test Interpretation	Post-Test Education and Follow-Up
Physician	17 (14.4)	25 (21.6)	21 (18.0)
Pharmacist	44 (37.3)	49 (42.2)	48 (41.0)
Nurse practitioner	8 (6.8)	5 (4.3)	8 (6.8)
Registered nurse	2 (1.7)	0 (0)	3 (2.6)
Genetic counsellor	33 (28.0)	27 (23.3)	23 (19.7)
More than one profession selected / those with knowledge and training regardless of profession	10 (8.5)	7 (6.0)	9 (7.7)
Unsure / it depends	3 (2.5)	2 (1.7)	4 (3.4)
None of the above	1 (0.9)	1 (0.9)	1 (0.9)
Total responses	118	116	117

**Table 4 pharmacy-10-00139-t004:** Top barriers identified by survey respondents (*n* = 117).

Barrier	% Selected *	% Selected as Top Barrier
Lack of health care providers knowledge	87.2	41.9
Cost / funding / reimbursement	81.2	29.1
Lack of guidelines / quality evidence	70.1	11.1
Lack of testing equipment	58.1	2.6
Lack of time to provide service	46.2	3.4
Delay in test results	40.2	3.4
Lack of electronic medical record integration	38.5	2.6
Ethical concerns	29.1	2.6
Legal concerns	28.2	0.9
Lack of patient acceptance	15.4	0.9
Social concerns	12.8	0
Unsure	1.7	1.7

* Participants could select more than one response.

## Data Availability

Not applicable.

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
