# Peer review of "Experience, Knowledge, and Perceptions of Pharmacogenomics among Pharmacists and Nurse Practitioners in Alberta Hospitals"

_pharmacy, 2022, doi:10.3390/pharmacy10060139_

Round 1

Reviewer 1 Report

Hayashi and Bousman conducted a survey to assess perceptions and knowledge of pharmacogenomics among health professionals in Alberta Hospital. The manuscript is well-written in general. The low response rate among the physician group was a concern, while it was acknowledged as a major limitation and the explanation seems to be reasonable. The authors also did a nice job by highlighting that “the results of this research are primarily representative of the knowledge for the latter two groups” (i.e., nurse practitioners and pharmacists). However, the reviewer wondered whether this important message should be reflected in the study title. Besides, the reviewer also wondered whether the study sample is representative of the population of nurse practitioners and pharmacists. It would be helpful to compare the demographics of survey respondents with that of the broader population in Alberta hospitals.

Additional comments were listed below for the authors’ reference to further improve this manuscript: 

1. Page 3, para 2: “There was no difference in prior education between those who had been practicing for more than 10 years compared to those who have been in practice for a shorter time (31.9% vs. 30.2%, respectively).” Please specify any statistical tests performed for group comparisons 

2. Page 4, para 1: Couldn’t locate Supplementary Materials Table S2. 

3. Page 4, figure 1: A minor suggestion is to modify the color of the bar charts in Figures B and D to match the color of pie slices for participants who have received PGx training or applied PGx in practice. 

4. Page 4-5, “Resource knowledge”: I suggest the authors modify this section to include more description of Figure 2. 

5. Page 5, figure 2: please consider adding percentage values in the bar graph to be consistent with other figures.

6. Figure 3, figure 3B: please confirm the asterisks on the Y axis and the missing notations if applicable. 

Author Response

Dear reviewer,

Thank you for taking the time to provide kind and insightful feedback that has improved the quality of our manuscript. Please see the noted revisions made per your advice. 

Reviewer 2 Report

Dear authors:
It was such a pleasure reviewing your well presented work.
The scope of this manuscript is extremely important, given the lack of information on the health professionals knowledge and experience in the field of pharmacogenomics, also extensive to clinical pharmaceutical practice.

The introduction adequately contextualizes the topic explored and the objectives are appropriately defined.

Also the methodology is adequate.

In the Results chapter, for the figures, when they have a legend, there is no added value in having a title, looking like duplicate information. Also in the figures, it is advisable to standardize the size of the legends (eg Figure 3).

As a conclusion, a suggestion for future research/work related to the topic explored could be included.

Author Response

(The authors gave the same response as above.)

Round 2

Reviewer 1 Report

Thank you for taking my suggestions into consideration. I do not have additional comments or concerns. Great work!